# Difference in racket head trajectory and muscle activity between the standard volley and the drop volley in tennis

**Ryosuke Furuya[1], Hikaru Yokoyama[2], Milos Dimic[1], Toshimasa Yanai[3], Tobias Vogt[4], Kazuyuki Kanosue[3]***

1 Graduate School of Sport Sciences, Waseda University, Tokorozawa, Saitama, Japan, 2 Department of Life Sciences, Graduate School of Arts and Sciences, The University of Tokyo, Meguro, Tokyo, Japan, 3 Faculty of Sport Sciences, Waseda University, Tokorozawa, Saitama, Japan, 4 Institute of Professional Sport Education and Sport Qualifications, German Sport University Cologne, Cologne, Germany

* kanosue@waseda.jp

## Abstract

Among tennis coaches and players, the standard volley and drop volley are considered basically similar, but muscles need to be relaxed (deactivation) just at the moment of impact when hitting the drop volley. However, this is not evidence-based. The aim of this study was to clarify racket head trajectory and muscle activity during the drop volley and to compare them with those of the standard volley. We hypothesized that 1) the racket head would move less forward for the drop volley than for the standard volley and 2) the wrist and elbow muscles be relaxed for the drop volley at the time of ball impact. Eleven male college students with sufficient tennis experience volunteered to participate in this study. Wireless EMG sensors recorded activation of the four arm muscles. Each subject performed the standard volley or the drop volley with both a forehand and a backhand from a position near the net. Four high speed video cameras (300 Hz) were set up on the court to measure ball speed and racket head trajectory. Returned ball speed of the drop volley was significantly lower than that of the standard volley ($p < 0.05$). The racket head moved less forward than in the standard volley, supporting the first hypothesis. Muscle activity of the drop volley, just before and after ball impact for both the forehand and backhand, was lower than that of the standard volley. However, the activity was in the form of a gradual increase as impact time approached, rather than a sudden deactivation (relaxation), which did not support the second hypothesis. For the drop volley, lower muscle activity in the forearm enabled a softer grip and thus allowed a "flip" movement of the racket to diminish the speed of the returned ball.

## Introduction

In tennis, when players are near or approaching the net, the shot they take most often involves a volley, which is a shot taken before the ball bounces. There are a number of volley types. The standard, and most used volley, is the standard volley. For this shot, the racket is moved forward, and slightly down. Forward racket movement is important, because it increases the

**Data Availability Statement:** All relevant data are within the manuscript.

**Funding:** The author(s) received no specific funding for this work.

**Competing interests:** The authors have declared that no competing interests exist.

velocity and control over the returned ball [1]. This makes it difficult for an opposing player to make a satisfactory response. For the standard volley, a greater increase in incoming ball speed requires a pronounced increase in forearm muscle activity. This amplifies grip force and wrist stiffness [2,3]. The skill involved in holding the grip tight and pushing the racket forward is important in the production of a good standard volley.

Another type of volley is the drop volley. It is often used when an opponent is anticipating a standard volley and is away or moving away from the net, or is behind the baseline. When using the drop volley, the intention is to slow down the speed of the returned ball as much as possible, and to drop the ball over but near the net and out of the opponent's reach. This shot starts with a racket position similar to that of the standard volley, but players have to absorb the energy of the incoming ball by utilizing the racket just like the movement that occurs when catching a raw egg with the hand [4]. Thus, the key for hitting an effective drop volley is loosening the grip just before ball impact [5]. This movement is very interesting from a biomechanical viewpoint as well as how motor control functions. From a coaching point of view, muscle relaxation is considered to be very important for accomplishing the delicate movements required for the drop volley. Since no study has been done so far on the muscle activity that occurs during the drop volley, the importance of muscle relaxation is just a general belief without any evidence, and it is ambiguous which muscles are involved and how much they should be relaxed. Also unknown is the actual racket movement during the production of a drop volley. The purpose of this study was to fill in this lack of knowledge, which would contribute not only to a basic understanding of the movement but also aid coaches in teaching how to execute the drop volley, since it is a very difficult shot for beginners. We compared the racket trajectories and muscle activities that occur during the production of a standard volley and a drop volley. We hypothesized that 1) the racket head would move less forward for the drop volley than for the standard volley, and 2) the wrist and elbow muscles would become relaxed at the time of ball impact for the drop volley. Muscle relaxation is generally considered as the absence or cessation of muscle activity, but in this study, to analyze it quantitatively in relation to the volley shots, we defined it as "a decrease in muscle activity as ball impact approached".

## Materials and methods

### Subjects

Eleven male tennis players (age: 20.8 ± 1.2 years; height: 173.3 ± 4.9 cm; weight: 65.0 ± 6.8 kg; time of tennis experience: 11.7 ± 2.2 years; average ± SD) volunteered to participate in this study. The drop volley is a shot that can only be played by players with sufficient experience and skill. Therefore, only players who declared that they utilized the drop volley with confidence, in actual games, were used as subjects. More specifically, five subjects were top level university tennis players at the Japanese national level, another five were top level university tennis players, and the remaining one had extensive experience as a recreational tennis player. One subject was left-handed and the others were right-handed. In accordance with the Declaration of Helsinki, the experiment procedure was explained to all subjects and each subject signed an informed consent. This study was approved by the Ethics committee of Waseda University. Subjects were told before the experiment that their photos might be used in scientific papers and they gave their consent to this. One subject was a minor (19 years). The study was non-invasive, and according to the ethical committee rule of Waseda University, to which both authors and subjects belong, when subjects are university students affiliated with Waseda University, parental consent is not required for them. This experiment was completed prior to the COVID-19 pandemic, and was conducted over a period of approximately two months starting in October 2018.

## Procedure

The experiment was conducted at a tennis court. In preparation for the incoming ball, each subject was asked to stand 2.5 m in front of the rear boundary of the service area, and to straddle the line dividing the right and left receiving areas. The subjects used their own rackets. A tape marker was attached to the top of the racket, and the subject hit balls propelled from a ball machine (Teniser, PM-100, SILVER REED, Japan) set at the center of the opposite side just behind the baseline (Fig 1). Each subject was told whether to hit the ball with a forehand or backhand, and to perform either a standard volley or a drop volley until 15 successful shots were completed. A shot was considered successful when the ball dropped inside the singles court on the opposite side. The speed of the projected ball was adjusted for each subject so that hitting both standard volleys and drop volleys was easy. Before the start of the experiment, the subject was given time to practice hitting the balls straight back to the ball machine with both standard and the drop volleys.

## Measurements

Four high-speed video cameras (gc-px1, JVC, Japan) were set on one side of the court such that the subject would face the camera at the moment of impact. The cameras were operated at 300 fps, and video data with the best image from the two cameras were used for analysis of racket head trajectory. Incoming and returned ball speeds as well as racket head movement were obtained from the video images (Fig 1). In order to obtain the time of impact between the racket and a ball, another two high-speed video cameras set at 1000 fps (fastec, TS3-S, FASTEC IMAGING, Japan) were positioned on the same side of the court as the other cameras. The impact of the ball on the racket was defined as the moment when the ball showed the greatest deformity. When trials were completed on one side, the cameras were moved to the opposite side so that, for the remaining forehand/backhand trials the subject would again be facing the camera. The players only performed single handed backhand volleys.

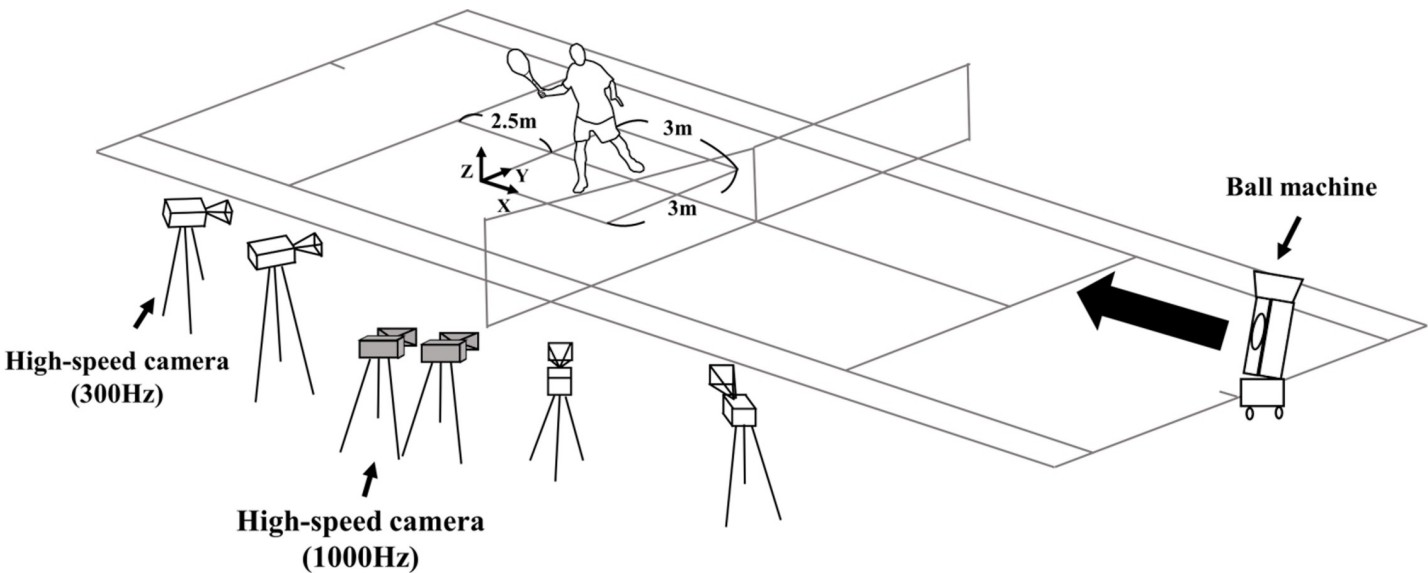

**Fig 1. Experimental design for forehand shots.** Video images of subject, racket and ball were captured with six cameras. Balls were projected from a machine. In the case of backhand shots all the cameras were set of the other side of the court.

## Electromyographic recordings

Electromyograms (EMGs) were recorded from four muscles of the dominant arm (flexor carpi radialis, FCR; extensor carpi radialis, ECR; biceps brachii, BB; triceps brachii, TB) with telemetry devices (Trigno wireless EMG system, Delsys, USA) supplying information to a data acquisition system (Powerlab, ADInstruments, Australia). Before attaching the devices, the skin was shaved and cleaned with alcohol to reduce the noise of the EMGs. The telemetry devices were placed on the FCR, ECR, and TB of each muscle belly in the manner of a previous study [2]. For the BB, it was attached to the center of muscle belly when the elbow was flexed. Before the experiment, it was confirmed that muscle activity was obtained from the appropriate muscle and that activity from the antagonist muscle was not included (no crosstalk). After the experiment session, subjects performed isometric maximum voluntary contraction (MVC) of each muscle for more than 5 s; maximal effort isometric contraction was performed as done in the previous study [6]. For each MVC, the average value for 3 s excluding the first second was calculated [6]. The trial was performed twice, and the average value was taken as the mean MVC value.

## Data analyses

To synchronize the video and EMG data, we used a synchronizer (PH-120, PH-140, DKH, Japan) that generates a trigger signal. The signal of the synchronizer was received by both the data acquisition system and video cameras. Sampling frequency for the EMG recordings was 2000Hz. EMG data were band-pass-filtered (20–450Hz, telemetry device property) and full wave rectified. The EMG data for each muscle were standardized as % EMG using the mean MVC value. The EMG data were low-pass filtered (zero lag fourth-order Butterworth at 4 Hz cutoff) and averaged over the 15 successful shots for each muscle.

To obtain the co-contraction level of forearm and upper arm muscles in the drop volley and the standard volley, the values of averaged muscle activities of the flexor muscles in each period relative to those of the extensor muscles were obtained, as:

$$\text{Co-contraction level} = \text{flexor muscle activity}/\text{extensor muscle activity} * 100\ (\%).$$

Racket head movement in three-dimensions was calculated using the Direct Linear Transformation algorithm [7]. The reference frame was as shown in Fig 1. The racket head was manually digitized from the video data of two of the four cameras sitting beside the subject. The analysis section was 30 ms before and after impact. Racket head trajectory was calculated by averaging the racket head displacement for each of the 15 trials in the X-Z plane (Fig 1). Coordinate values at impact were defined as the origin, and were subtracted from each coordinate value in the period from 30 ms before to 30 ms after the impact. This reference was defined as the racket reference frame. The origin of the racket reference frame was the impact point and the X axis was directed horizontally forward toward the net. Ball speed was obtained as the average of the period ranging from 50 ms prior to and after impact [8]. Subsequently, the ratio between the incoming and returned ball speeds (Percent ball speed) were calculated. Percent ball speed was specified as:

$$\text{Percent ball speed} = \text{returned ball speed}/\text{incoming ball speed} * 100\ (\%)$$

## Statistical analyses

One subject was excluded due to a defect in the camera used for the calculation of the trajectory of the racket head. However, the camera used to detect the impact did work. Thus, the EMG analysis was able to be done on all 11 subjects. Data are presented as the mean ± standard deviation.

Ball speed was compared using paired t-tests with SPSS software (IBM SPSS Statistic ver. 25.0 for windows, IBM, USA) and evaluated with $p < 0.05$ as the level of significance. Statistical parametric mapping (SPM) techniques [9] were used to compare the differences of the EMG amplitude and racket head trajectory between the standard volley and the drop volley. SPM analysis can find statistically different timing portions in continuous time-series data between conditions rather than compare discrete time points [10]. We performed SPM analysis for time series data on each muscle and each direction of the racket head trajectory (X and Z axes corresponding to horizontal and vertical directions, respectively). All the SPM analyses were performed with SPM1D toolbox available for Matlab (version M.0.4.7) [10]. Normal distribution was initially checked with a normality test implemented in SPM1D. Because the normality was not shown for all the datasets, we used SPM analysis of non-parametric paired t-tests. The output of SPM analysis provides SPM [11] values for each time point of the time-series data and the threshold corresponding to the set alpha level. The alpha level was altered via the Sidak correction for multiple comparisons (alpha = 0.0253 for racket head trajectory [two directions] and co-contraction level [two muscle pairs], and alpha = 0.0127 for individual muscle activity [four muscles]). If the SPM{t} values exceed the threshold (indicated as red dash lines in Figs 4, 7 and 8), there are significant differences between the conditions in the time series.

## Results

### Ball speed

Paired t-test revealed that ball speed of the drop volley was significantly lower than that of the standard volley. This was true for each subject in both the forehand and backhand (Fig 2, $p < 0.001$). Averaged percent ball speeds (returned/incoming) of the standard volley was about 100% (forehand; 104 ± 9%, backhand; 101 ± 10%), but for the drop volley it was only about 40% (forehand; 39 ± 4%, backhand; 40 ± 3%).

### Racket head trajectory

Fig 3 shows a series of photographs of the racket trajectories during the standard volley and the drop volley of one subject. The standard volley involved a forward movement of the racket before impact, whereas in the drop volley the racket did not move forward but was rather pushed backward due to the ball impact. Fig 4A shows averaged racket head trajectories during the standard volley and the drop volley for 10 subjects (one subject was excluded due to the defect of a camera for the calculation of the trajectory of the racket head. However, the camera used to detect the impact worked. Thus, the analysis of EMG was able to be done in all 11 subjects). The graph shows the trajectory from 30 ms before to 30 ms after impact. The racket head was displaced forward to a greater extent in the standard volley than in the drop volley. In the standard volley, the forward motion of the racket was damped at the moment of impact and the racket moved vertically downward along the Z axis after impact. In the case of the drop volley, the racket head moved forward only slightly before ball impact, and it moved backward after impact, especially for the forehand drop volley. In other words, the racket was pushed back due to the impact. On the other hand, the racket head trajectories in the vertical direction were similar in both the standard and drop volleys.

The SPM results showed that the X axis displacement was significantly different between the standard and drop volleys for the entire time period from 30 ms before to 30 ms after impact except the impact timing ($p < 0.05$) for both forehand and backhand shots (Fig 4B). This supports the above-mentioned explanation. On the other hand, for the Z axis, a significant difference between the standard and drop volley was observed only approximately 20 ms to 30 ms after impact for the forehand shot ($p < 0.05$, Fig 4C).

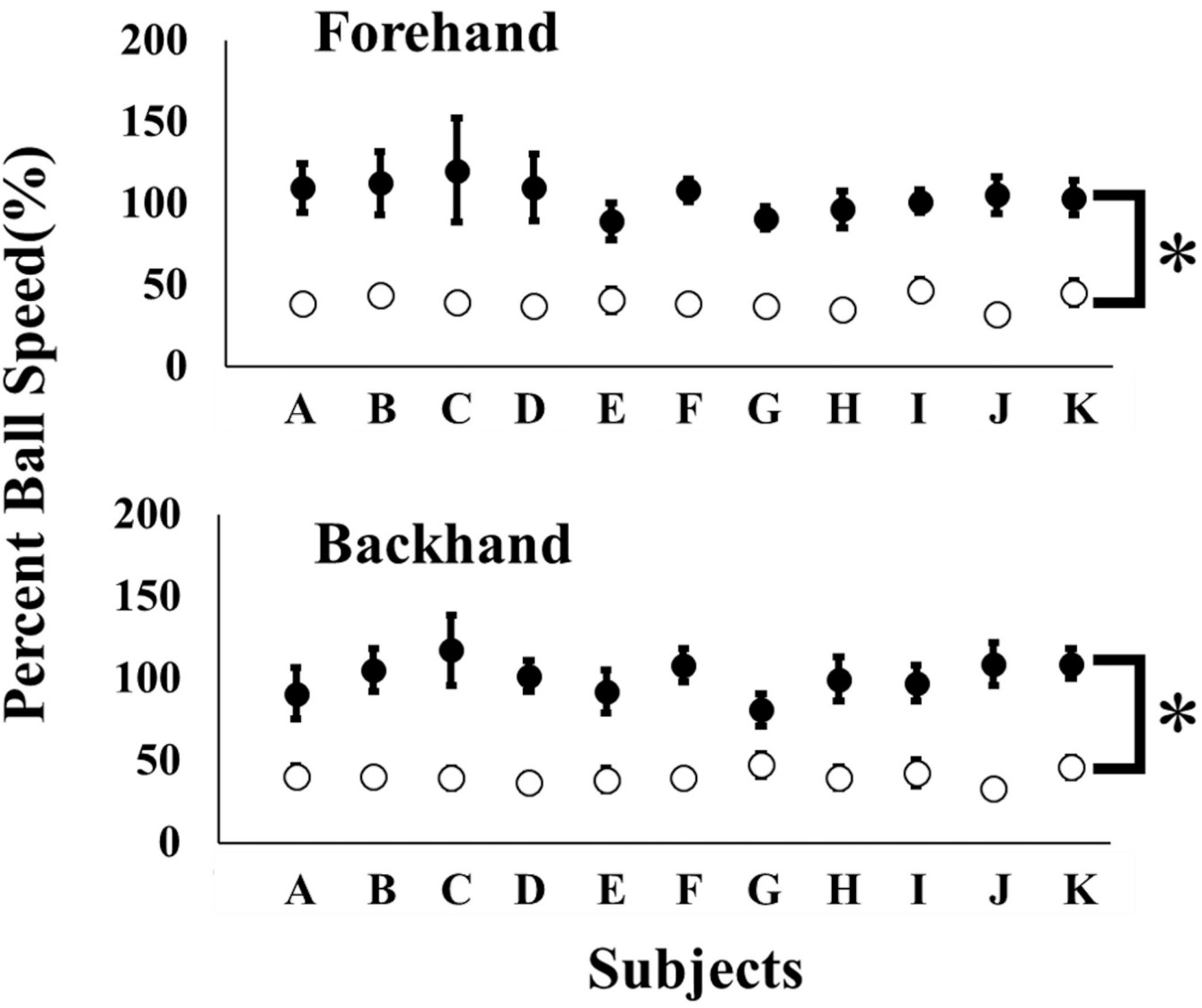

**Fig 2. Percent ball speed.** The percent ball speed of the drop volley is significantly lower than the standard volley in all subjects (A~K, * $p < 0.05$). Open and closed circles donated the drop volley and the standard volley, respectively.

The x-component of racket head velocity in the period 30 ms before impact was greater for the backhand drop volley (3 ± 1.8m/s) than for the forehand drop volley (1.1 ± 1.1 m/s) ($p < 0.05$), but there was no difference between the forehand and backhand standard volley.

## Muscle activity

Figs 5 and 6 display examples of EMGs when hitting the standard volley and drop volley with the forehand (Fig 5) and backhand (Fig 6) in one and the same subject. The muscle activity during drop volley is generally smaller than that during standard volley.

Group averaged EMGs of the standard volley and drop volley are shown in Figs 7 (forehand) and 8 (backhand). Overall, regardless of the type of shot (standard/drop volley), forehand/backhand, forearm/ upper arm, or flexor/extensor, muscle activity tended to gradually increase toward, or slightly after, impact.

**Standard volley**

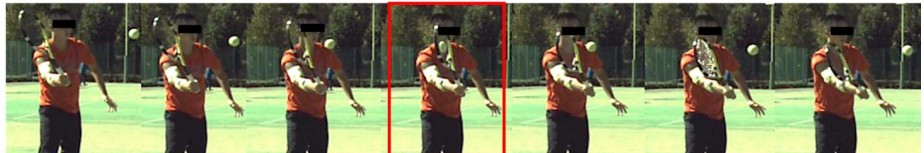

**Drop volley**

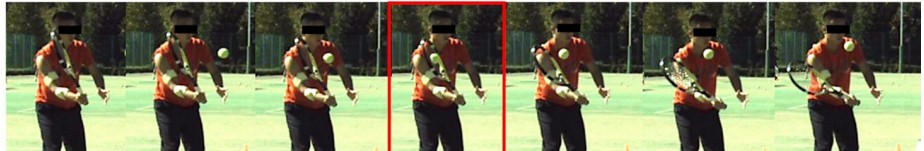

**Fig 3. A series of photographs of the racket trajectories in the standard volley and the drop volley in one subject.** Red border denoted impact. The pictures surrounded with red are the ball impact.

**Forehand shots (Fig 7).** Regarding FCR and ECR activity, there were no significant differences between the volley types. In BB, activity during the standard volley was greater in the period approximately 20 ms to 50 ms after impact as compared to that during the drop volley (p < 0.05). For TB, activity during the standard volley was also greater at impact than that during the drop volley (p < 0.05).

**Backhand shots (Fig 8).** FCR activity did not show any significant differences between the volley types. ECR activity in the period from approximately 30 ms to 10 ms before the impact was greater during the standard volley than during the drop volley (p < 0.05). In BB, the activity from just after the impact to 50 ms after the impact was greater during the standard volley compared to that during drop volley (p < 0.05). In TB, the activity in the period from 100 ms before to approximately 10 ms after impact was greater during the standard volley as compared to that during drop volley (p < 0.05).

**Co-contraction level.** The co-contraction level of forearm muscles and upper arm muscles are presented in the bottom portions of Figs 7 and 8. For the forearm, the co-contraction levels for the drop volley were around 100% (blue lines) for both the forehand and backhand shot. On the other hand, levels of the standard volley were above 100% for the forehand (Fig 7), and below 100% for the backhand (Fig 8). SPM results for the forehand shot showed that the co-contraction level in the period from approximately 10 ms before to 20 ms after impact were significantly higher during the standard volley than during the drop volley (p < 0.05, Fig 7). For the backhand shot, there was no significant difference between the two volley types (Fig 8).

For the upper arm, the co-contraction level showed no obvious change with volley type or with forehand/backhand. The only significant difference was found in the co-contraction level from approximately 25 ms to 10 ms before impact, which was greater during the drop volley than during the standard volley (p < 0.05, Fig 7). For the backhand shot, there were no significant differences in the co-contraction level between the two volley types (Fig 8).

## Discussion

The purpose of this study was to analyze racket movement and muscle activity in the drop volley as compared with those of the standard volley. We hypothesized that 1) the racket head would move less forward for the drop volley than the standard volley, and 2) the wrist and elbow muscles would be relaxed at the time of ball impact for the drop volley.

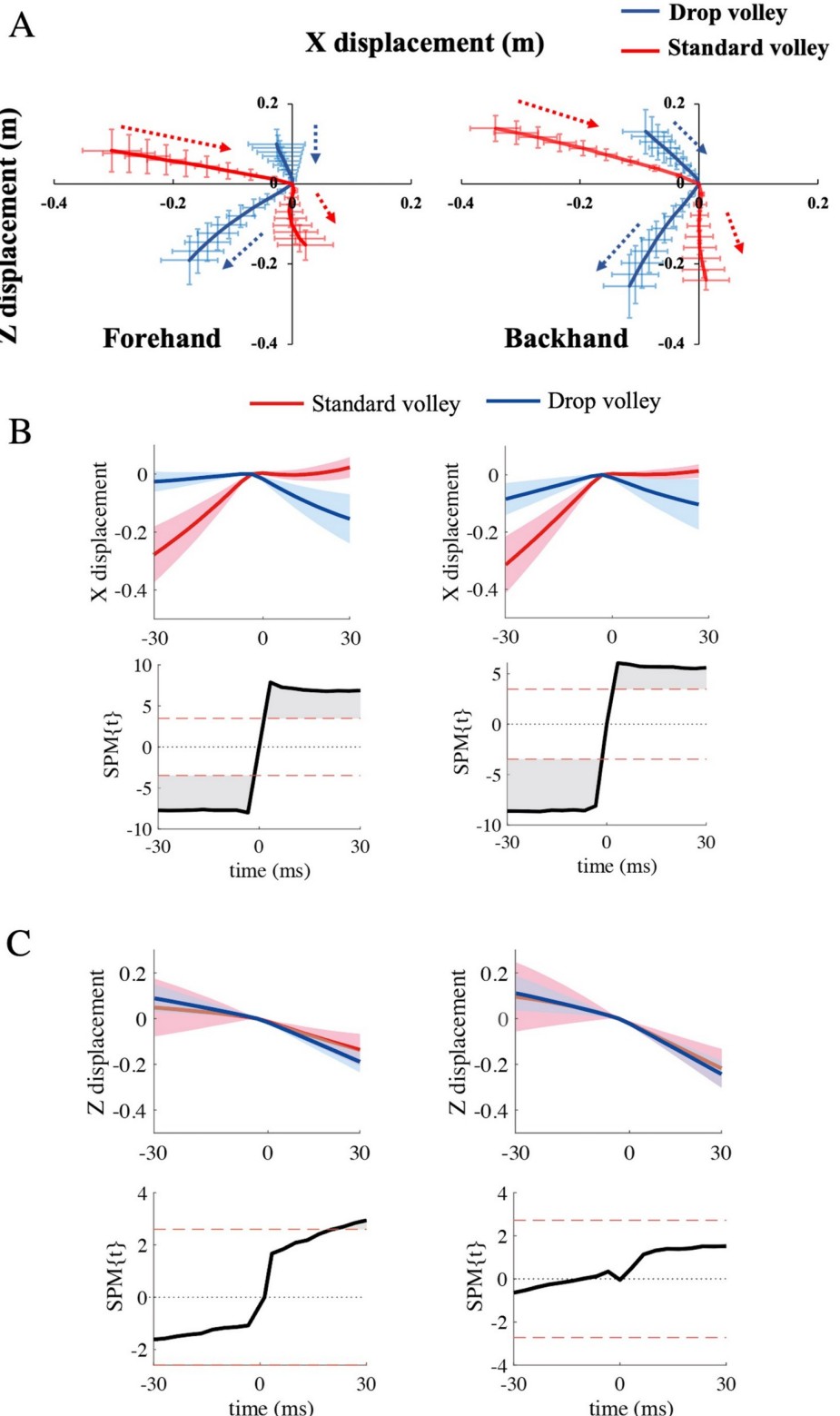

**Fig 4.** The averaged racket head trajectory for forehand (left) and backhand (right). (A) 2D view of the averaged racket head trajectory. Blue lines display the drop volley and red lines display the standard volley; the origins denote ball impact. (B) Averaged time series of X displacement of racket head trajectory obtained with an SPM analysis. Blue and red lines indicate the drop volley and standard volley, respectively. Gray areas indicate timing when SPM{t} values

exceeded the alpha level threshold, which is displayed as a red dashed lines. (C) Averaged time series of Z displacement of the racket head trajectory obtained with an SPM analysis.

When employing the standard volley, players need to hit the ball strongly in order to maximize the speed of the returned ball. Racket speed is an essential factor for increasing speed of the returned ball. Translational racket movement, rather than a rotational one, is required (Fig 3), because the standard volley is often played in a temporally-restricted situation.

In contrast, for the drop volley, racket head displacement was smaller before impact as compared to the standard volley, and the racket was flipped back after impact (Figs 3 and 4A). This finding did support the first working hypothesis. Indeed, the racket head trajectory of the X axis for the drop volley was significantly smaller than that of the standard volley (Fig 4A and 4B). This movement led to a significantly lower speed of ball return than with the standard volley (Fig 2). This could have been accomplished with a decreased racket head speed in the forward direction in the same period as was seen for the standard volley, indicating that it is not necessary for the drop volley to move the racket forward.

A previous study on forehand groundstroke shots showed that FCR and ECR activity increase as the impact moment is approached [12]. Further, when hitting the single-handed backhand stroke, ECR muscle activity also gradually increases as the impact time is approached [13]. Such muscle activity is likely important for wrist function during the swing of the racket. The standard volleys are also expected to require a certain amount of muscle activity in order to accelerate the racket. When hitting the standard volley, the wrist should be locked and the racket being gripped tightly so that the forces of the racket and arm be integrated. Chow et al. [2] reported that in the standard volley tennis players tighten their grip shortly before impact and this tightness lasts even after impact. FCR and ECR likely have roles as stabilizers during forehand and backhand volleys, respectively. Indeed, FCR in the forehand standard volley increased activity toward the moment of impact (Fig 7). Co-contraction level also increased and reached near 300% around impact time (Fig 7), indicating that FCR worked dominantly over ECR. To the contrary, for the backhand standard volley, ECR increased activity through the swing (Fig 8), with a co-contraction level below 100% (Fig 8). This indicates an ECR dominance. These muscle activity would generate the torque needed to counteract the wrist flexion/extension caused by the impact. As for the upper arm muscles in the standard volley, it is especially interesting that TB was dominantly activated over BB, with a co-contraction level below 100%, both for the forehand and backhand shots in the period prior to the impact (Figs 7 and 8). This likely occurred because the standard volley relies on a pushing movement involving elbow extension in both the forehand and backhand shots. To achieve this movement of the upper arm, BB should not activated. This avoids a co-contraction.

For the drop volley, FCR activity of the forehand shot tended to be lower as compared to that of the standard volley (around the time of impact) (Fig 7). Conversely, for the backhand shot, ECR activity was lower for the drop volley as compared to the standard volley (Fig 8). This corresponds well to the fact that in the drop volley, racket head displacement was smaller before impact as compared to that of the standard volley because there is no need to counteract ball momentum. This is because the racket was flipped back after impact (Figs 3 and 4). We initially hypothesized that the wrist and elbow muscles would be relaxed at the time of ball impact for the drop volley. In the present study, we defined "relaxation" in relation to the drop volley as "a decrease in activity as ball impact approached". However, it was commonly observed that at ball impact there was no relaxation in the activity of any muscle, and rather a gradual increase in activity as impact approached (Figs 7 and 8); that is, the EMGs never decreased toward ball impact. This result did not support the hypothesis. Actually, for the

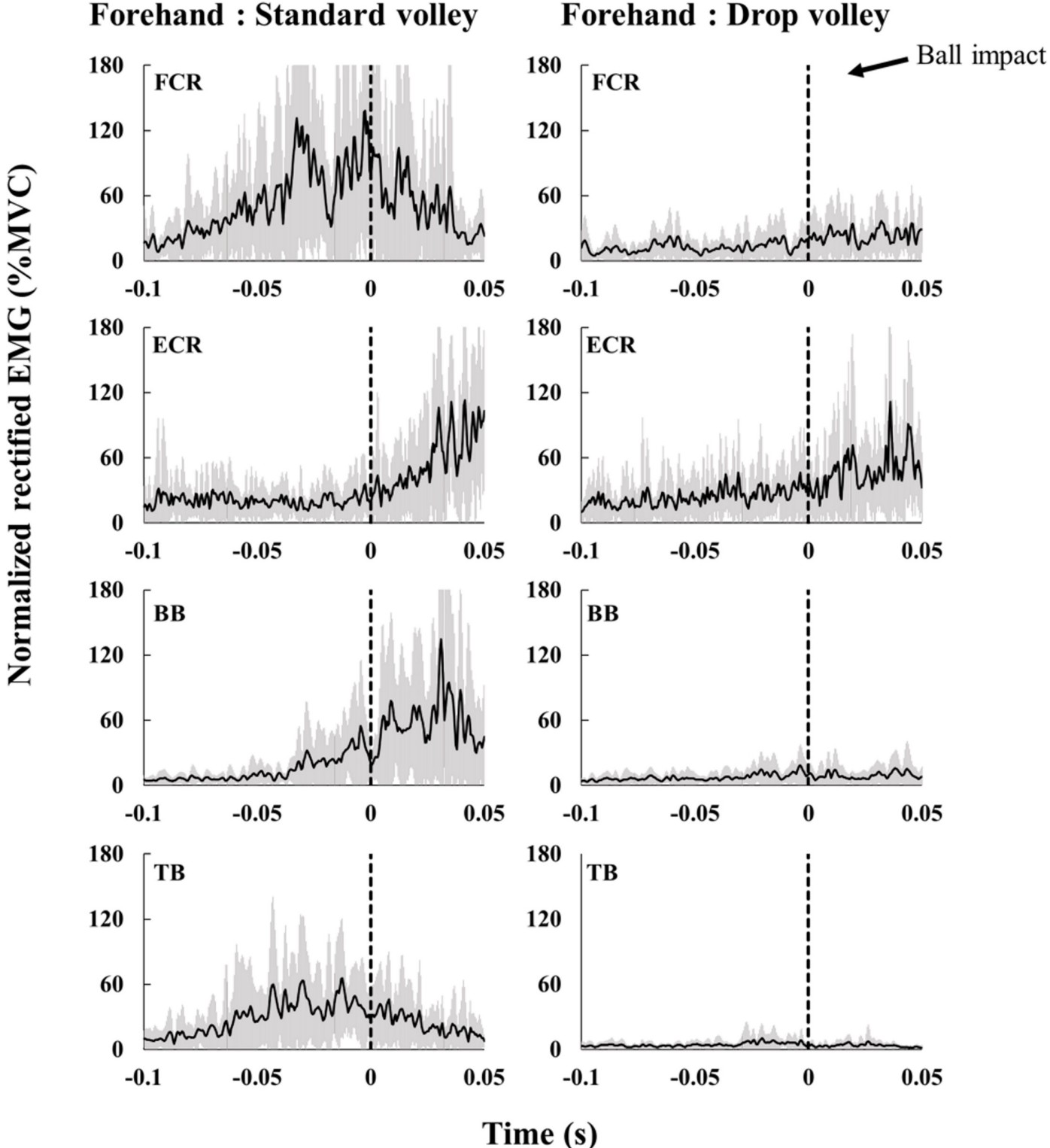

**Fig 5.** Rectified EMGs of one representative subject for the standard forehand volley (left) and the forehand drop volley (right). Black lines indicate averaged values of all trials, grey areas indicate standard deviation. The dashed line (0 s) denotes the impact.

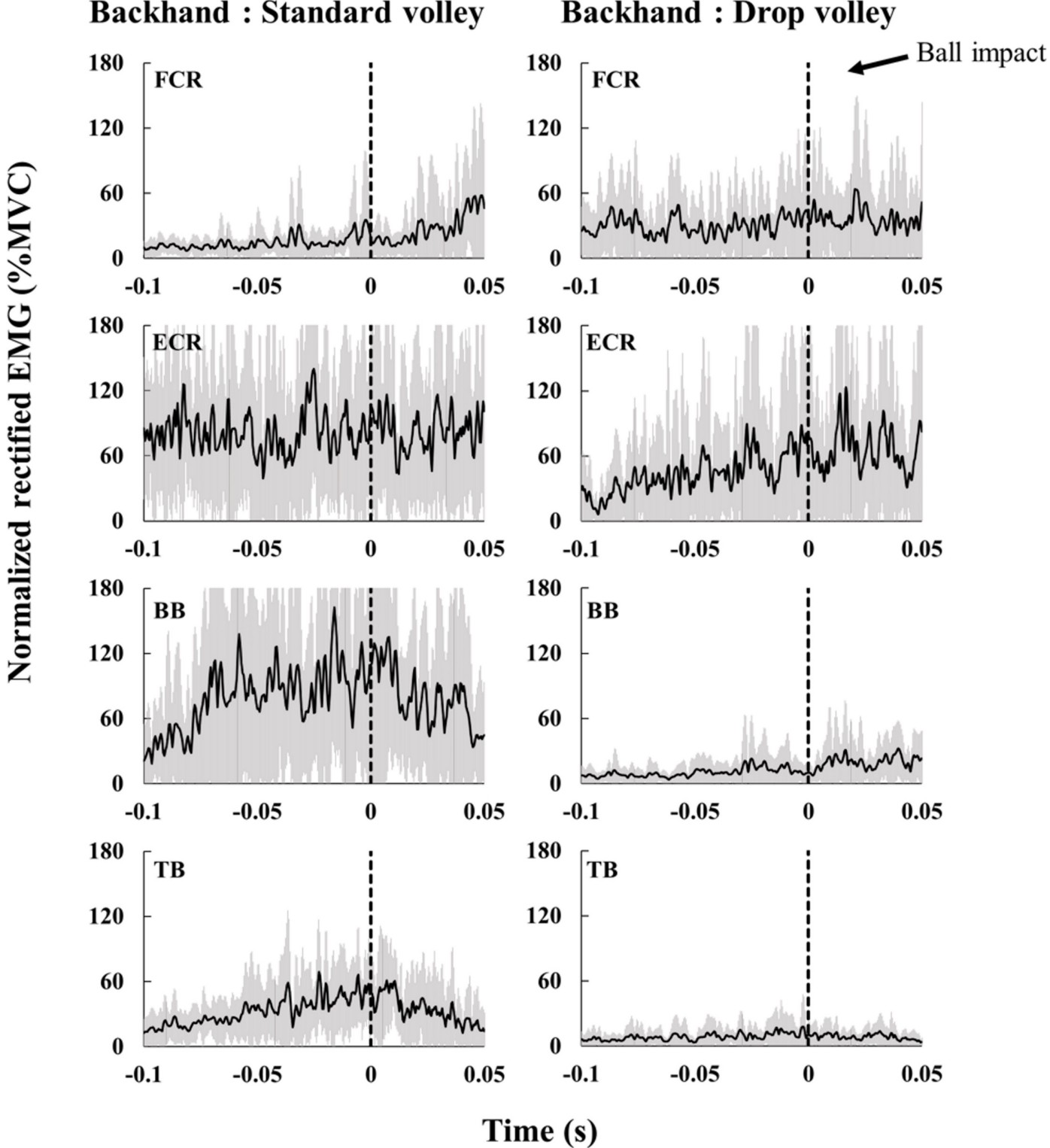

**Fig 6.** Rectified EMGs of one representative subject for the standard backhand volley (left) and the backhand drop volley (right). Black lines indicate averaged values of all trials, grey areas indicate standard deviation. The dashed line (0 s) denotes the impact.

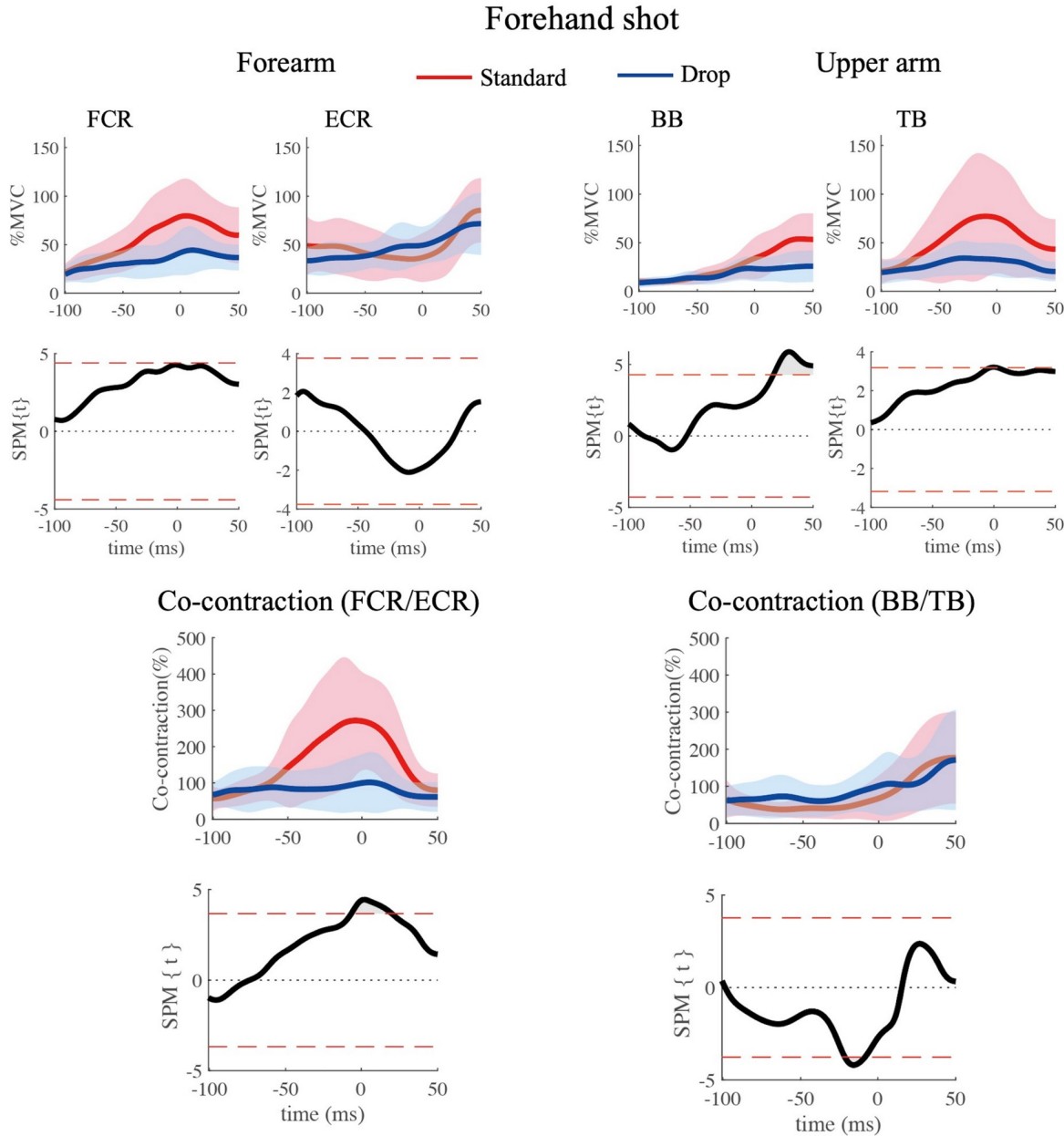

**Fig 7. Four muscle activity; comparing the standard volley (red lines) and the drop volley (blue lines) for the forehand shot.** Averaged time series data arw presented with corresponding results of the SPM analysis. Gray areas indicate the times when SPM{t} values exceeded the alpha level threshold, which is displayed as red dash lines. Time 0 depicts the time of impact.

drop volley, just absorbing the energy of the incoming ball is not enough to be an efficient shot, and the speed and the direction of the returned ball need to be controlled such that the ball passes over the net, but falls as close to the net as possible. Therefore, it is necessary to lock the wrist and maintain the grip to some extent. This is likely the reason why the forearm muscles were not completely relaxed in the drop volley and showed activation, although weakly in comparison with the standard volley. In addition, the co-contraction level of the forearm is close to 100% in the drop volley, for both the forehand and backhand. In the case of the drop volley, there is no need to create a strong impact momentum. The co-contraction with lower

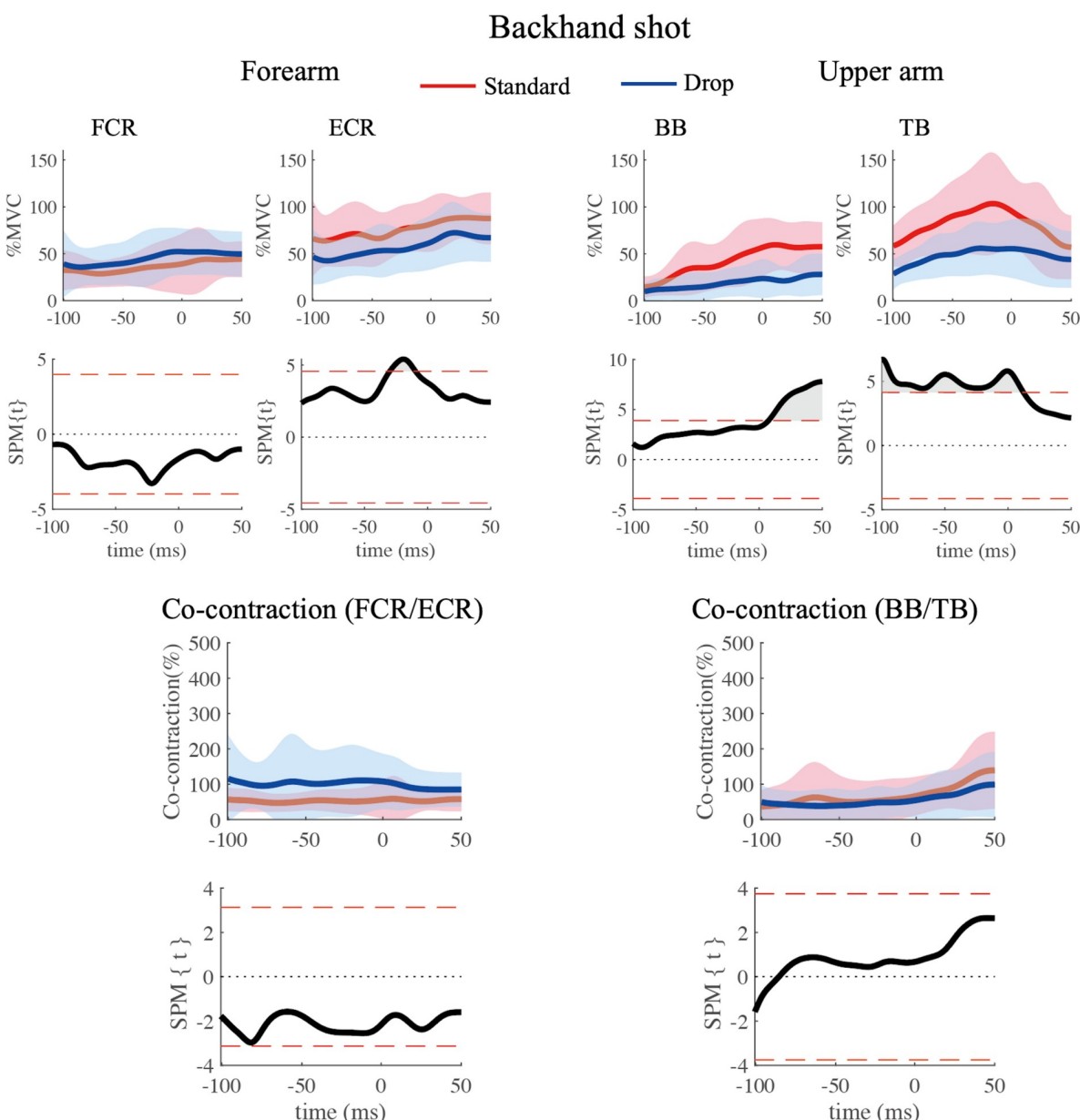

**Fig 8. Four muscle activity comparing the standard volley (red lines) and the drop volley (blue lines) for the backhand shot.** Averaged time series data were presented with corresponding results of the SPM analysis. Gray areas indicate the times when SPM{t} values exceeded the alpha level threshold, which is displayed as red dash lines. Time 0 depicts of the time of impact.

activity in the forearm muscles, therefore, produces a soft grip that facilitates the flipped back movement which occurs at the moment of impact in the drop volley. This generates a lower momentum of the ball when it leaves the racket.

In a manner similar to that of the forearm muscles, upper arm muscle activity in both TB and BB during the drop volley was generally weaker than the standard volley (Figs 7 and 8). This would contribute to absorbing the energy of the incoming ball [4]. The drop volley requires a flip movement, so stabilization of the elbow and shoulder joints is not required as compared to the standard volley.

The results of the present study don't necessarily correspond to conventional teaching theory which holds that "the drop volley should be performed by loosening the racket grip just before the impact", as noted in the introduction. The drop volley is actually executed with a low, but definite level of muscle activity, rather than with muscle relaxation. To hit a drop volley, the forearm and upper muscles should exert a weaker level of activation; enough to hold the racket as impact approaches, not relaxing those muscles, and moving the racket less than for the standard volley. Thus, ball speed needs to be dampened, but still be fast enough so that the ball goes over the net but not to go so far that the opponent can easily hit it. To accomplish this players have to master the appropriate degree of force with which to grip the racket (muscle contraction), which depends on the speed of the coming ball. It is also important to not push the racket as far forward as for the standard volley. These suggestions increase the possibility of making a successful shot.

The drop volley is a difficult shot, and only high level players can properly do it and utilize it as a tactic in a real match. Although we tried to recruit as many subjects as possible who satisfied the condition, we could find only eleven. Therefore, there is a possibility that the present findings could apply only to a limited range of players.

## Conclusion

This study analyzed muscle activity and racket head trajectory of the drop volley in comparison with the standard volley. A drop volley's function is to return the ball at a much slower speed than that of the standard volley. We hypothesized that 1) the racket head would move less forward for the drop volley than for the standard volley, and 2) the wrist and elbow muscles would be relaxed at the time of ball impact for the drop volley. The racket head moved less forward for the drop volley than the standard volley, supporting the first hypothesis. In the drop volley, after ball impact the racket head is flipped back. Forearm and upper arm EMG recordings during a drop volley showed an overall lower activity level than did those of the standard volley, but the activities were never in the form of sudden decrease at impact, and rather, a gradual increase as impact approached. This did not support the second hypothesis. The results of this study contradict the general notion in the conventional coaching method of how to hit the drop volley, which is to relax muscles at the time of impact.

## Acknowledgments

The authors thank Dr. Larry Crawshaw for English editing, and the tennis players who volunteered to participate in this study as well as their coaches for allowing usage of the data.

## Author Contributions

**Data curation:** Hikaru Yokoyama.

**Formal analysis:** Ryosuke Furuya, Milos Dimic.

**Funding acquisition:** Kazuyuki Kanosue.

**Methodology:** Hikaru Yokoyama, Toshimasa Yanai, Tobias Vogt.

**Project administration:** Kazuyuki Kanosue.

**Resources:** Toshimasa Yanai, Kazuyuki Kanosue.

**Supervision:** Toshimasa Yanai, Tobias Vogt, Kazuyuki Kanosue.

**Visualization:** Ryosuke Furuya, Milos Dimic.

**Writing – original draft:** Ryosuke Furuya, Milos Dimic.

**Writing – review & editing:** Hikaru Yokoyama, Toshimasa Yanai, Tobias Vogt, Kazuyuki Kanosue.

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
