## [Decision Letter · Decision Letter 0]

29 Apr 2021

PONE-D-20-39873

Difference in muscle activity between the standard volley and the drop volley in tennis.

PLOS ONE

Dear Dr. Kanosue,

Thank you for submitting your manuscript to PLOS ONE. After careful consideration, we feel that it has merit but does not fully meet PLOS ONE’s publication criteria as it currently stands. Therefore, we invite you to submit a revised version of the manuscript that addresses the points raised during the review process.

Since there is a conflict between reviewer's recommendations, enhanced statistical analysis and English-revising will be necessary for acceptance of the revised manuscript. Please submit your revised manuscript by Jun 13 2021 11:59PM. If you will need more time than this to complete your revisions, please reply to this message or contact the journal office at plosone@plos.org. Please include the following items when submitting your revised manuscript:

We look forward to receiving your revised manuscript.

Kind regards,

Hyojung Choo

Academic Editor

PLOS ONE

Journal Requirements:

2. Please clarify whether you have written consent for publication for the participant’s picture in the figure. For further information please refer to our policy on informed consent for publication https://journals.plos.org/plosone/s/human-subjects-research#loc-Patient-Privacy-and-Informed-Consent-for-Publication;
https://journals.plos.org/plosone/s/file?id=8ce6/plos-consent-form-english.pdf

In your Methods section, please provide additional information about the participant recruitment method and the demographic details of your participants. Please ensure you have provided sufficient details to replicate the analyses such as: a) the recruitment date range (month and year), b) a description of any inclusion/exclusion criteria that were applied to participant recruitment.

3. Please clarify whether any minors were included in your study, please ensure you have also stated whether you obtained consent from parents or guardians of the minors included in the study or whether the research ethics committee or IRB specifically waived the need for their consent.

Reviewers' comments:

Reviewer's Responses to Questions

**Comments to the Author**

1. Is the manuscript technically sound, and do the data support the conclusions?

Reviewer #1: Yes

Reviewer #2: Partly

2. Has the statistical analysis been performed appropriately and rigorously? 

Reviewer #1: Yes

Reviewer #2: No

3. Have the authors made all data underlying the findings in their manuscript fully available?

Reviewer #1: Yes

Reviewer #2: Yes

4. Is the manuscript presented in an intelligible fashion and written in standard English?

Reviewer #1: Yes

Reviewer #2: No

5. Review Comments to the Author

Reviewer #1: GENERAL COMMENT:

The purpose of this study was to describe and compare muscle activity and racket head trajectory in the tennis drop volley and in the standard volley. Authors monitored surface EMG activity of four muscles acting on elbow (2) and wrist joints (2), and six high-speed video cameras were used to calculate racket head trajectory, ball speed and the time of impact between the racket and a ball. This is a study where the authors have clearly invested significant time into data collection and analysis. This study would be of value to readers who train tennis athletes or to researchers concerned with neuromuscular patterns in different sport skills. I feel the manuscript could be considerably improved and in my opinion a major revision is necessary in order to accept the paper. Please consider the comments listed below.

SPECIFIC COMMENTS:

Introduction

Lines 62-65: I strongly suggest the authors to reformulate the Hypophesis. In the abstract the objective of the study was defined as “The aim of this study was to clarify muscle activity and racket head trajectory in the tennis drop volley and compare it with that of the most often used the standard volley” (Lines 27-28). In the end of the Introduction the authors wrote “We compared the racket trajectories and muscle activities that occur during the production of a standard volley and a drop volley” (Lines 62-32). So I don’t understand why there was no hypothesis relatively to comparation of EMG muscle activity between both types of volley. Even because a considerable part of the Discussion and Conclusions was dedicated to this EMG comparation.

Concerning the first hypothesis (Lines 63-64) “muscles were relaxed at the time of ball impact for a drop volley”: what do the authors mean with "muscles were relaxed"? What was the quantitative criterion (for example EMG threshold level) to define relaxation? Notice that in conclusions chapter and in the abstract there are no comments about how the study answered to this hypothesis.

Materials and Methods

Line 69. How was the sample size calculated?

Lines 112-113. I have concerns about the high probability of crosstalk contamination in the EMG signals of flexor carpi radialis (FCR) and extensor carpi radialis (ECR) muscles. Can the authors guarantee that the EMG signals collected are exactly from these muscles? Since the electrodes diameter, the spacing between the electrodes and the accurate electrodes location on the surface of the muscle are the three most influential factors that contribute to the amount of crosstalk (DeLuca et al., 2011, J Biomechanics 4, 555-561), it would be important to provide specific information in the Materials and Methods about these aspects. On the other hand, wouldn't it be better not to be so specific in identifying the monitored muscles? It would not be more correct to say that the EMG signals were collected, for example, from wrist flexors and wrist extensors?

Line 114. Please explain how were the MVCs performed for each of the four muscles. This description is necessary to better understand the percentage level of muscle activation during the volley and to compare it with the relative values obtained in other studies.

Lines 123-124. If I understood well, the EMG signals of each subject and type of volley “were averaged over the 15 successful shots for each muscle”. I suppose that these 15 trials had different durations. How was this solved? Were the EMG signals submitted to a time normalization process?

Line 227. “one subject was excluded due to camera malfunction”. The EMG signals of this subject were used? If yes, how was the instant of ball impact determined?

Results

A table with average values and standard deviation of the different variables in each subject could be very useful to understand the group results and the variability between different subjects.

For example, how do the authors explain that the Biceps Brachii shows before impact in the backhand standard volley, average EMG values under ±60% MVC (it is impossible to see the precise value based on the graph of Figure 6) when the EMG of the subject represented in Figure 4, in the same volley and phase, shows EMG values higher than 150%MVC? And, exception to the beginning, the EMG values of BB before impact were always clearly greater than 60%MVC?

If we compare the BB and TB values of activation before impact during the backhand standard voley, in the individual values (Figure 4) the BB shows much higher values than the TB, but when we look to the average values (Figure 6) the trend is opposite with the TB presenting higher values (as it would be expected since it is the main agonist of elbow extension that is performed before ball contact, Line 286).

If this is a “representative subject”, as you say in the legend of the Figure 4 (Lines 171-173) please explain the apparent contradiction between the average results and the results of this subject.

Discussion

Lines 271-273. “This explains why muscle activity in the standard volley is greater before impact, especially that of the flexor muscle for the forehand volley and that of the extensor for the backhand volley.”

I do not find in the Results chapter data that support this finding. And it also doesn't seem coherent to the averaged values of both muscles we can observe before and after impact in Figures 5 and 6.

Line 275. The co-contraction pattern is very important to joint stabilization, specially to stabilize the wrist joint during impact ball impact. Considering the different strategies of each type of volley, probably this is a variable that differs between standard and drop volley. Why the authors did not consider the possibility of quantifying the co-contraction level in each joint and compare it between standard and drop volley?

Lines 284-287. The Triceps Brachii results are discussed here. But I did not find any discussion or analysis about the Biceps Brachii results, one of the four muscles that were studied. I suggest you dedicate in the Discussion some attention to this muscle, commented on the EMG results, specifically the differences between the two types of voley and your interpretation for these differences. It would be also helpful to relate the EMG patterns of both muscles (BB and TB).

References

It is necessary to standardize the references according the PLOS ONE rules. Some examples:

In Reference 1 the Authors Names are UPPERCASE while in the other references they are in SENTENCE CASE.

In some references the Name of the Journal it is written in full (Reference 1 and 10) while in other references Name of the Journal is abbreviated.

In References 4 and 5 pages are missing.

In Reference 6 volume and number are missing.

MINOR ISSUES

Line 88. Please standardize the terms. Replace “with both regular and the drop volleys” with “with both standard and the drop volleys”.

Line 129. Add “.” after “Transformation algorithm (7)”.

Line 155. Add “.” after “(forehand; 39 ± 4%, backhand; 40 ± 3%)”.

Line 159. Add “.” after “respectively”.

Line 214. Please clarify if the values of Figure 5 are average values from all subjects.

Line 218. Please clarify if the values of Figure 6 are average values from all subjects.

Line 267. I suggest eliminating “for the standard volley” since it is repeated in the sentence.

Line 288. Would the authors want to say “Overall EMG magnitude for the forearm muscles” instead of “Overall EMG magnitude for the arm muscles”? Because all the paragraph is dedicated to discussion of forearm muscles activation and its role in the wrist joint.

Reviewer #2: General Comments:

This paper examines the muscle activation patterns and 3 D kinematics of "standard" and drop volley strokes in tennis on a sample of 11 male national level and recreation level tennis players. The manuscript is well structured and the study has potential practical relevance. There are however various aspects that require considerable revision and/ or reworking as described in the specific comments below:

Specific Comments:

The Abstract and Introduction are not very well written and require significant correction of grammar and syntax throughout. There is a tendency for verbosity and passive sentence structures that make the text difficult to read.

ln 29: "To record... for arm muscles" it may be better to use a more direct expression for example: "Wireless EMG sensors recorded muscle activations of four arm muscles". Similarly ln 32: "To measure......(300 Hz)." perhaps revise to: "Four high speed video cameras (300 Hz) were set up on the court to measure ball speed and racket head trajectory".

There are numerous instances of this type of syntax throughout the Abstract, Introduction and Discussion, therefore a sentence by sentence revision is required.

The introduction needs to establish a clear justification (i.e. rationale ) for the study. The current justification is not sufficient, the text needs to explain to the reader why the study is needed and/or worthwhile.

The sample size is very small and the ability range is relatively large and this presents a limiting factor for the study. The data acquisition techniques appear to be sound however the data analysis (statistical) methods do not seem to be ideal for analyzing the EMG and 3D kinematic time series data and this is a matter of some concern.

The data analysis focuses on comparing EMG time series and muscle activations and this appeared to involve various averaging procedures on the time series data overall and then on a section by section basis. I do not think this is an optimal approach and the authors should perhaps rethink how they should analyze their data. The outcome that results from the analysis is (for example) that the overall ECR muscle comparison between standard volley and drop volley was not significant but the section by section analysis was significant. There are other more appropriate methods for comparing mean time series data such as Statistical Parametric Mapping (SPM) as well as other techniques.

Finally, the overall idea and key hypothesis of the study was that muscle relaxation would be very important in the drop volley and I am not sure if this is a good assumption. The drop volley will require movements of the racket that generate a lower momentum of the ball when it leaves the racket. I don't think this necessarily requires the muscles to relax, but rather to control the movement of the racket head in such a way that the racket ensures the ball speed is lower. This can be achieved through various muscle activation strategies, with relaxation being only one of various possible muscle activation strategies.

6. PLOS authors have the option to publish the peer review history of their article (what does this mean?). If published, this will include your full peer review and any attached files.

Reviewer #1: No

Reviewer #2: No

---

## [Author Response · Author response to Decision Letter 0]

20 Jul 2021

Reviewer #1: GENERAL COMMENT:

The purpose of this study was to describe and compare muscle activity and racket head trajectory in the tennis drop volley and in the standard volley. Authors monitored surface EMG activity of four muscles acting on elbow (2) and wrist joints (2), and six high-speed video cameras were used to calculate racket head trajectory, ball speed and the time of impact between the racket and a ball. This is a study where the authors have clearly invested significant time into data collection and analysis. This study would be of value to readers who train tennis athletes or to researchers concerned with neuromuscular patterns in different sport skills. I feel the manuscript could be considerably improved and in my opinion a major revision is necessary in order to accept the paper. Please consider the comments listed below.

A. We greatly appreciate the reviewer for the positive evaluation of this study, and especially the deep understanding the challenges involved in doing the experiments. We tried to follow the reviewers suggestions as much as possible and, thus, we hope that the manuscript has been satisfactorily improved.

SPECIFIC COMMENTS:

Introduction

Lines 62-65: I strongly suggest the authors to reformulate the Hypophesis. In the abstract the objective of the study was defined as “The aim of this study was to clarify muscle activity and racket head trajectory in the tennis drop volley and compare it with that of the most often used the standard volley” (Lines 27-28). In the end of the Introduction the authors wrote “We compared the racket trajectories and muscle activities that occur during the production of a standard volley and a drop volley” (Lines 62-32). So I don’t understand why there was no hypothesis relatively to comparation of EMG muscle activity between both types of volley. Even because a considerable part of the Discussion and Conclusions was dedicated to this EMG comparation.

Concerning the first hypothesis (Lines 63-64) “muscles were relaxed at the time of ball impact for a drop volley”: what do the authors mean with "muscles were relaxed"? What was the quantitative criterion (for example EMG threshold level) to define relaxation? Notice that in conclusions chapter and in the abstract there are no comments about how the study answered to this hypothesis.

A. We appreciate the positive evaluation of our study and constructive suggestions. In the revised manuscript we have clearly stated the hypotheses as: “We hypothesized that 1) the racket head would move less forward for the drop volley than for the standard volley and 2) the wrist and elbow muscles be relaxed for the drop volley at the time of ball impact “ The abstract and conclusion have been accordingly revised, mentioning the previously stated hypotheses: (abstract; Lines; 31-33, conclusion; Lines; 394-396). 

In the introduction, the previous research on EMG in the standard volley was mentioned. For the drop volley, there is no previous research involving an EMG, but in terms of coaching it is generally considered that “relaxation” is necessary for the drop volley. As for “relaxation”, we tried to describe its analysis in a logical way, that is:

1) 

Muscle relaxation is generally considered as the absence or cessation of muscle activity, but in this study, to analyze it quantitatively in relation to the volley shots, we defined it as “a decrease in muscle activity as ball impact approached” (Lines; 77-78 ),

2) we utilized three time bins of 50 ms, before and after the impact (Line 150-152), and, 

3) we judged whether a muscle was “relaxed” at the moment of ball impact by whether there was a significant decrease in muscle activity in section 2 (just before the impact) or section 3 (just after the impact) as compared to the activity present in the section 1 (100 to 50 ms prior to the impact) 

4) Interestingly, in no case was this criterion satisfied, so we could conclude that there was no “relaxation” in the drop volley, as is generally considered to occur and coached as such.

 We hope that now the logic of the whole paper has become clearer, and the paper is rational and understandable.

Materials and Methods

Line 69. How was the sample size calculated?

A. The number of subjects, eleven, was not determined with the calculation of the sample size, but rather by the limit of how many players we could recruit. The drop volley is a difficult shot and only high level players can properly execute and utilize it in real games. Therefore, the subjects pool was limited to players who declared that they had utilized the drop volley with confidence in actual games. Although we tried to recruit as many subjects as possible who satisfied this condition, we could find only eleven. Therefore, there is a possibility that the present findings could apply to only a limited range of players. We added this recruiting limitation in more detail in the Methods Section (Lines; 85-88) and also added the limitation to the Discussion (Lines; 386-389).

Lines 112-113. I have concerns about the high probability of crosstalk contamination in the EMG signals of flexor carpi radialis (FCR) and extensor carpi radialis (ECR) muscles. Can the authors guarantee that the EMG signals collected are exactly from these muscles? Since the electrodes diameter, the spacing between the electrodes and the accurate electrodes location on the surface of the muscle are the three most influential factors that contribute to the amount of crosstalk (DeLuca et al., 2011, J Biomechanics 4, 555-561), it would be important to provide specific information in the Materials and Methods about these aspects. On the other hand, wouldn't it be better not to be so specific in identifying the monitored muscles? It would not be more correct to say that the EMG signals were collected, for example, from wrist flexors and wrist extensors?

A. We appreciate the comments. The sensor was attached to the subject in accordance with a previous study (Chow et al., 2007). We added this information to the method section (Lines;134-135). The following is the EMG activity during recording of the MVC data (not shown in the main manuscript). This figure clearly shows that the cross talk, if any, was relatively small so that the EMG could appropriately be recorded from the target muscles (see the word file). 

Line 114. Please explain how were the MVCs performed for each of the four muscles. This description is necessary to better understand the percentage level of muscle activation during the volley and to compare it with the relative values obtained in other studies.

A. We added information concerning the method about the how MVC was collected, so it now reads as: “subjects performed isometric maximum voluntary contraction (MVC) of each muscle for more than 5 s; maximal effort isometric contraction was performed as done in our previous study (6). “ (Lines; 138-140) 

Lines 123-124. If I understood well, the EMG signals of each subject and type of volley “were averaged over the 15 successful shots for each muscle”. I suppose that these 15 trials had different durations. How was this solved? Were the EMG signals submitted to a time normalization process?

A. The impact time between ball and racket is only 3ms. Of course the activation period of each muscle was much longer than this and differed among muscles. Analysis of this study was focused on the short period around the time of impact. So, we used the absolute time from the impact without normalization. We added this explanation in the Method Section, which now reads as: “Since the time of impact between the racket and ball is very short (about 3 ms), and considerably shorter than the duration of muscle activity, the analysis was conducted based on the absolute time from the moment of impact. “ (Lines;152-154).

Line 227. “one subject was excluded due to camera malfunction”. The EMG signals of this subject were used? If yes, how was the instant of ball impact determined?

A. The camera which captured racket head motion failed on this trial, but other cameras detecting the impact worked. This information was added in the Result Section of racket head trajectory, which now reads as: “One subject was excluded due to a defect in the camera used for the calculation of the trajectory of the racket head. However, the camera used to detect the impact did work. Thus, the EMG analysis was able to be done on all 11 subjects. “ (Line;178-179).

Results

A table with average values and standard deviation of the different variables in each subject could be very useful to understand the group results and the variability between different subjects.

A. We appreciate the valuable suggestion. Accordingly, we made tables showing the means of muscle activity in each subject for forehand/backhand and standard/drop volley (shown below). Although they only contain means (and not SD) for each subject, they contain too much information and are complicated. Therefore, we decided not to use them in the revised manuscript. However, we tried to make the manuscript as logical and straightforward as possible, and we hope that the important points of this study can be conveyed without the tables. Thanks for the suggestion (see the word file).. 

For example, how do the authors explain that the Biceps Brachii shows before impact in the backhand standard volley, average EMG values under ±60% MVC (it is impossible to see the precise value based on the graph of Figure 6) when the EMG of the subject represented in Figure 4, in the same volley and phase, shows EMG values higher than 150%MVC? And, exception to the beginning, the EMG values of BB before impact were always clearly greater than 60%MVC? 

If we compare the BB and TB values of activation before impact during the backhand standard volley, in the individual values (Figure 4) the BB shows much higher values than the TB, but when we look to the average values (Figure 6) the trend is opposite with the TB presenting higher values (as it would be expected since it is the main agonist of elbow extension that is performed before ball contact, Line 286).If this is a “representative subject”, as you say in the legend of the Figure 4 (Lines 171-173) please explain the apparent contradiction between the average results and the results of this subject.

A. We appreciate the detailed examination of our data. One of the main findings of this study is that EMG activity was weak in the drop volley as compared to the standard volley. We chose Figure 6 because it clearly shows this characteristic. We omitted “representative”, because, as the reviewer pointed out, the example of the figure does not necessarily match all the average tendencies shown in Figure 8. 

EMG activities have large individual variation, probably because the action differed from one subject to another. This might be the reason why “in the individual values (Figure 6) the BB shows much higher values than the TB, but when we look at the average values (Figure 8) the trend is in the opposite direction, with TB presenting higher values.

Discussion

Lines 271-273. “This explains why muscle activity in the standard volley is greater before impact, especially that of the flexor muscle for the forehand volley and that of the extensor for the backhand volley.”

I do not find in the Results chapter data that support this finding. And it also doesn't seem coherent to the averaged values of both muscles we can observe before and after impact in Figures 5 and 6.

A. We appreciate the reviewer’s comments. The explanation of the old manuscript was not sufficient. To show the temporal EMG changes more clearly, we deleted the old Figs. 5 and 6, in which the time bins (25 ms) were too short. Instead, we made analysis of EMGs for 50ms bins before and after impact and showed them as the new Figs. 7 and 8, which also included the data of co-contraction levels as the reviewer suggested. Accordingly, we were now able to clearly show the importance of FCR and ECR in the forehand and backhand standard volley, respectively. We have modified the corresponding Discussion part, as “FCR and ECR likely have roles as stabilizers during forehand and backhand volleys, respectively. Indeed, FCR in the forehand standard volley increased activity around the moment of impact (sections 2 and 3 in Fig 7) with a co-contraction level of over 100 %, and co-contraction level was also higher around impact (the bottom portion of Fig. 7), indicating that FCR worked dominantly over ECR. To the contrary, for the backhand standard volley, ECR activity increased activity toward impact (sections 2 and 3 in Fig 8) with the co-contraction level below 100 % (the bottom portion of Fig 8), indicating ECR’s dominance. This muscle activity would generate the torque needed to counteract the wrist flexion/extension caused by the impact.” (Lines; 337-344). 

Line 275. The co-contraction pattern is very important to joint stabilization, specially to stabilize the wrist joint during impact ball impact. Considering the different strategies of each type of volley, probably this is a variable that differs between standard and drop volley. Why the authors did not consider the possibility of quantifying the co-contraction level in each joint and compare it between standard and drop volley?

A. We appreciate the important suggestion. We added new figures concerning co-contraction level in the new figures at the bottom portion of Fig. 7 (forehand) and 8 (backhand) (lines; 278~293). Furthermore, related discussion was added in the discussions of the standard volley (Lines; 338-349) and drop volley (Lines; 364-369)

Lines 284-287. The Triceps Brachii results are discussed here. But I did not find any discussion or analysis about the Biceps Brachii results, one of the four muscles that were studied. I suggest you dedicate in the Discussion some attention to this muscle, commented on the EMG results, specifically the differences between the two types of volley and your interpretation for these differences. It would be also helpful to relate the EMG patterns of both muscles (BB and TB).

A. Thanks for the suggestion. We added discussion concerning BB activity and the BB/TB relationship, so it now reads as: “In a manner similar to that of the forearm muscles, upper arm muscle activity in both TB and BB during the drop volley was generally weaker than the standard volley (Figs 7 and 8). This would contribute to absorbing the energy of the incoming ball (4). The drop volley requires a flip movement, so stabilization of the elbow and shoulder joints is not required as compared to the standard volley. ” (Lines; 370-374) 

References 

It is necessary to standardize the references according the PLOS ONE rules. Some examples:

In Reference 1 the Authors Names are UPPERCASE while in the other references they are in SENTENCE CASE.

In some references the Name of the Journal it is written in full (Reference 1 and 10) while in other references Name of the Journal is abbreviated.

In References 4 and 5 pages are missing.

In Reference 6 volume and number are missing.

A. We appreciate the detailed checking. We have corrected all the references. Reference 6 has been just accepted‘, and the volume in which the paper appears is not informed. 

MINOR ISSUES

Line 88. Please standardize the terms. Replace “with both regular and the drop volleys” with “with both standard and the drop volleys”.

A. We appreciate the detailed checking. We have corrected that sentence as suggested.

Line 129. Add “.” after “Transformation algorithm (7)”.

A. We appreciate the detailed checking. We have corrected that sentence as suggested. 

Line 155. Add “.” after “(forehand; 39 ± 4%, backhand; 40 ± 3%)”.

A. We appreciate the detailed checking. We have corrected that sentence as suggested.

Line 159. Add “.” after “respectively”.

A. We appreciate the detailed checking. We have corrected that sentence as suggested.

Line 214. Please clarify if the values of Figure 5 are average values from all subjects.

Line 218. Please clarify if the values of Figure 6 are average values from all subjects.

A. We appreciate the detailed checking. We have added the required sentence (Lines; 232-233).

Line 267. I suggest eliminating “for the standard volley” since it is repeated in the sentence.

Line 288. Would the authors want to say “Overall EMG magnitude for the forearm muscles” instead of “Overall EMG magnitude for the arm muscles”? Because all the paragraph is dedicated to discussion of forearm muscles activation and its role in the wrist joint.

A. We appreciate the detailed checking. We have corrected all the noted problems. 

 

Reviewer #2: General Comments:

This paper examines the muscle activation patterns and 3 D kinematics of "standard" and drop volley strokes in tennis on a sample of 11 male national level and recreation level tennis players. The manuscript is well structured and the study has potential practical relevance. There are however various aspects that require considerable revision and/ or reworking as described in the specific comments below:

A. We appreciate the positive evaluation. We tried to revise the manuscript according to the reviewer’s suggestions. We hope it is now appropriately improved.

Specific Comments:

The Abstract and Introduction are not very well written and require significant correction of grammar and syntax throughout. There is a tendency for verbosity and passive sentence structures that make the text difficult to read.

ln 29: "To record... for arm muscles" it may be better to use a more direct expression for example: "Wireless EMG sensors recorded muscle activations of four arm muscles". Similarly ln 32: "To measure......(300 Hz)." perhaps revise to: "Four high speed video cameras (300 Hz) were set up on the court to measure ball speed and racket head trajectory".

There are numerous instances of this type of syntax throughout the Abstract, Introduction and Discussion, therefore a sentence by sentence revision is required.

A. We appreciate the suggestion. The revised manuscript was checked by a physiologist who is an English native-speaker.

The introduction needs to establish a clear justification (i.e. rationale ) for the study. The current justification is not sufficient, the text needs to explain to the reader why the study is needed and/or worthwhile.

A. We appreciate the important suggestion. We modified the Introduction so that the purpose and rational of this study are now straightforward. 

B. We stated two definite hypotheses.

The sample size is very small and the ability range is relatively large and this presents a limiting factor for the study. The data acquisition techniques appear to be sound however the data analysis (statistical) methods do not seem to be ideal for analyzing the EMG and 3D kinematic time series data and this is a matter of some concern. 

A. Thanks for the important suggestion. The present experiment was done in eleven subjects. The drop volley is a difficult shot and only players of high level can properly do it as a tactic in a real match. We added an explanation of how we recruited subjects, so, we added: “The drop volley is a shot that can only be played by players with sufficient experience and skill. Therefore, only players who declared that they utilized the drop volley with confidence, in actual games, were used as subjects.” (Lines; 84-86) Although we have tried to recruit as many subjects as possible who satisfied this condition, we could find only eleven. Therefore, there is a possibility that the present findings could apply to only a limited range of players. We added this limitation in the Discussion (Lines; 386-389).

As for the EMG, we have done additional analysis, and tried to describe it in a more logical way, that is,

1) although relaxation is generally considered as the cessation or absence of muscular activity, in this study we defined it as “a decrease in muscle activity as ball impact approached” (Line: 78),

2) we set three time bins of 50 ms each, before and after impact (Lines; 150-152), and, 

3) we judged a muscle was “relaxed”, at the moment of ball impact, if there was a significant decrease in muscle activity in section 2 (just before the impact) or section 3 (just after the impact) as compared to that in section 1 (100 to 50 ms prior to the impact) (Lines; 157-169 ). 

4) Interestingly, in no case was this criterion satisfied, so that we could conclude that there was no “relaxation” in the drop volley, as is generally considered to occur and coached as such.

The data analysis focuses on comparing EMG time series and muscle activations and this appeared to involve various averaging procedures on the time series data overall and then on a section by section basis. I do not think this is an optimal approach and the authors should perhaps rethink how they should analyze their data. The outcome that results from the analysis is (for example) that the overall ECR muscle comparison between standard volley and drop volley was not significant but the section by section analysis was significant. There are other more appropriate methods for comparing mean time series data such as Statistical Parametric Mapping (SPM) as well as other techniques.

A. We appreciate the very important suggestion. The EMG was averaged for the 25 ms intervals from 100 ms before the impact to 50 ms after the impact during the standard volley and the drop volley. We now feel that this interval was too small. And in the revised manuscript we focused more on muscle “relaxation” at the time of impact, and tried to analyze it more quantitatively, as noted above. We hope the way we did this was correct and helped us pick up important facts underlying the drop volley.

Finally, the overall idea and key hypothesis of the study was that muscle relaxation would be very important in the drop volley and I am not sure if this is a good assumption. The drop volley will require movements of the racket that generate a lower momentum of the ball when it leaves the racket. I don't think this necessarily requires the muscles to relax, but rather to control the movement of the racket head in such a way that the racket ensures the ball speed is lower. This can be achieved through various muscle activation strategies, with relaxation being only one of various possible muscle activation strategies.

A. This is an important point. From a coaching point of view, the emphasis has been generally made on muscle relaxation, but the results of this study, rather, showed that muscle activity increased towards the time of impact, and co-contraction was also observed, although activity level was low as compared with the standard volley. Furthermore, the trajectory of the racket is smaller than that of the standard volley. Therefore, gripping the racket handle softly would facilitate the flipped back movement and generate a lower momentum of the ball. However, the grip should be appropriate so that the returned ball speed is enough to go over the net but not to go too far so that the opponent easily hit it. Therefore, once coaches knew the present result, they should teach players to hold the grip softly and not relax. We have modified Discussion part concerned so that it now reads, as “Actually, for the drop volley, just absorbing the energy of the incoming ball is not enough to be an efficient shot, and the speed and the direction of the returned ball need to be controlled such that the ball passes over the net, but falls as close to the net as possible. Therefore, it is necessary to lock the wrist and maintain the grip to some extent. This is likely the reason why the forearm muscles were not completely relaxed in the drop volley and showed activation, although weakly in comparison with the standard volley.” (lines; 359-364,). “Muscle relaxation” is a working hypothesis, so we left it as it was.

---

## [Decision Letter · Decision Letter 1]

9 Aug 2021

PONE-D-20-39873R1

Difference in racket head trajectory and muscle activity between the standard volley and the drop volley in tennis.

PLOS ONE

Dear Dr. Kanosue,

Thank you for submitting your manuscript to PLOS ONE. After careful consideration, we feel that it has merit but does not fully meet PLOS ONE’s publication criteria as it currently stands. Therefore, we invite you to submit a revised version of the manuscript that addresses the points raised during the review process.

Please clearly address the concerns from reviewer 2 regarding analysis methods. Please revise analysis methods using an established method for time series/ curve data analysis or justify not using the established method. This response will help us to decide the acceptance of this manuscript. 

We look forward to receiving your revised manuscript.

Kind regards,

Hyojung Choo

Academic Editor

PLOS ONE

Journal Requirements:

Reviewers' comments:

Reviewer's Responses to Questions

**Comments to the Author**

1. If the authors have adequately addressed your comments raised in a previous round of review and you feel that this manuscript is now acceptable for publication, you may indicate that here to bypass the “Comments to the Author” section, enter your conflict of interest statement in the “Confidential to Editor” section, and submit your "Accept" recommendation.

Reviewer #1: All comments have been addressed

Reviewer #2: (No Response)

2. Is the manuscript technically sound, and do the data support the conclusions?

Reviewer #1: Yes

Reviewer #2: No

3. Has the statistical analysis been performed appropriately and rigorously? 

Reviewer #1: Yes

Reviewer #2: No

4. Have the authors made all data underlying the findings in their manuscript fully available?

Reviewer #1: Yes

Reviewer #2: No

5. Is the manuscript presented in an intelligible fashion and written in standard English?

Reviewer #1: Yes

Reviewer #2: Yes

6. Review Comments to the Author

Reviewer #1: Authors answered positively to my previous comments and included the purposed changes in the manuscript.

I thank the authors for having responded positively to my comments. Congratulations for the great work.

Reviewer #2: The authors have completed substantial revisions to this manuscript and have addressed several of the significant concerns I identified in my first review. The clarity of writing and grammar are much improved and the methods are much clearer.

The most important concerns I identified previously related to the appropriateness of the analysis methods when examining time series data and the effects of the various averaging processes which could compound the analysis. I suggested that these concerns may be addressed using established methods for time series analysis such as functional data analysis or statistical parametric mapping (random field theory). The authors elected to revise the analysis but only changed the time intervals for their analysis. The difficulties of averaging data from within subjects (trials and interval averaging) and pooling this with between subjects data to derive a global within and between subjects averages still remains. Furthermore the authors did not use an established method for time series/ curve data analysis to examine the data and this presents further problems.

I believe the authors may have some valuable data that could provide some useful insights but we cannot really be sure of this until the data are appropriately analysed. This was the major concern I raised and I do not believe it has been adequately addressed.

7. PLOS authors have the option to publish the peer review history of their article (what does this mean?). If published, this will include your full peer review and any attached files.

Reviewer #1: No

Reviewer #2: No

---

## [Author Response · Author response to Decision Letter 1]

26 Aug 2021

Dear editor and reviewers

I submitted it in the separated files (Cover letter and Response to reviewers). 

I appreciated to cooperation.

---

## [Editor Report · Decision Letter 2]

31 Aug 2021

Difference in racket head trajectory and muscle activity between the standard volley and the drop volley in tennis.

PONE-D-20-39873R2

Dear Dr. Kanosue,

We’re pleased to inform you that your manuscript has been judged scientifically suitable for publication and will be formally accepted for publication once it meets all outstanding technical requirements.

Kind regards,

Hyojung Choo

Academic Editor

PLOS ONE
---

## [Editor Report · Acceptance letter]

6 Sep 2021

PONE-D-20-39873R2 

Difference in racket head trajectory and muscle activity between the standard volley and the drop volley in tennis. 

Dear Dr. Kanosue:

I'm pleased to inform you that your manuscript has been deemed suitable for publication in PLOS ONE. Congratulations! Your manuscript is now with our production department. 

Kind regards, 

on behalf of

Dr. Hyojung Choo 

Academic Editor

PLOS ONE